# Revisiting Bacterial Ubiquitin Ligase Effectors: Weapons for Host Exploitation

**DOI:** 10.3390/ijms19113576

**Published:** 2018-11-13

**Authors:** Antonio Pisano, Francesco Albano, Eleonora Vecchio, Maurizio Renna, Giuseppe Scala, Ileana Quinto, Giuseppe Fiume

**Affiliations:** 1Department of Experimental and Clinical Medicine, University “Magna Graecia” of Catanzaro, 88100 Catanzaro, Italy; albano@unicz.it (F.A.); eleonoravecchio@unicz.it (E.V.); scala@unicz.it (G.S.); quinto@unicz.it (I.Q.); 2Department of Molecular Medicine and Medical Biotechnologies, University of Naples “Federico II”, 80131 Naples, Italy; maurizio.renna@unina.it

**Keywords:** ubiquitin ligase, T3SS, T4SS

## Abstract

Protein ubiquitylation plays a central role in eukaryotic cell physiology. It is involved in several regulatory processes, ranging from protein folding or degradation, subcellular localization of proteins, vesicular trafficking and endocytosis to DNA repair, cell cycle, innate immunity, autophagy, and apoptosis. As such, it is reasonable that pathogens have developed a way to exploit such a crucial system to enhance their virulence against the host. Hence, bacteria have evolved a wide range of effectors capable of mimicking the main players of the eukaryotic ubiquitin system, in particular ubiquitin ligases, by interfering with host physiology. Here, we give an overview of this topic and, in particular, we detail and discuss the mechanisms developed by pathogenic bacteria to hijack the host ubiquitination system for their own benefit.

## 1. Introduction

Ubiquitylation is a post-translational protein modification that plays a pivotal role in regulating cellular homeostasis [1]. It starts with the attachment of ubiquitin, a small globular protein of 76 amino acids to the target protein substrates by covalent binding of the carboxy-terminal of ubiquitin to the epsilon amino-group of a lysine of the substrate [2]. This process is iterative as a second ubiquitin moiety is bound to the carboxy-terminal of the first one through covalent bound to an epsilon amino-group of a specific lysine contained in the ubiquitin itself—including K6, K11, K27, K29, K33, K48, or K63—giving rise to very long ubiquitin chains [3,4]. Besides the canonical ubiquitylation, involving the covalent binding of the carboxy-terminal of ubiquitin to the epsilon amino-group of a lysine residue within the substrate protein, ubiquitylation at non-canonical, lysine-independent sites can also occur in cells. Non-canonical ubiquitylation sites include the free amine of methionine N-terminus of the polypeptide backbone, as well as cysteine thiol and hydroxyl groups contained in serine, threonine, and tyrosine [5]. Among the proteins ubiquitylated at N-terminus, notable examples are MyoD, Erk, p21, ARF, the HPV protein HPV16 and the Epstein–Barr viral proteins LMP1 and LMP2 [5]. Paradigmatic examples of cysteine-ubiquitylated proteins include the peroxisomal proteins Pex5p and Pex20p [6]. The proteins that are ubiquitylated at serine, threonine and tyrosine via hydroxyester linkages, include the T-cell antigen receptor alpha, that when fails to assemble with the other chains, it undergoes to endoplasmic reticulum-associated degradation (ERAD), a process requiring firstly the retro-translocation of the protein from the ER, its ubiquitylation on either serine or threonine residues, which eventually allow its proteasomal degradation [7,8]; likewise, the pro-apoptotic protein Bid, upon unconventional ubiquitylation, is degraded via the proteasome; the major histocompatibility complex I (MHC-I) heavy chain can undergo to serine- or threonine-ubiquitylation by the viral E3 ligase mK3 and, thus, targeted for ERAD [5]. It is interesting to note how this non-canonical ubiquitylation followed by ERAD is frequently observed during viral infections. Among those, specific examples are represented by the non-canonical ubiquitylation and degradation of host proteins BST-2, CD4, and NS-1, which are mediated by specific interactions of the HIV protein Vpu with the SCFβ-TRCP E3 ligase [5].

The ubiquitylation reaction cascade is guaranteed by the coordinated and sequential action of three enzymes, namely: E1, the activating enzyme, E2, the conjugating enzyme, and E3, the ligase enzyme [9]. In particular, the glycine carboxy-terminal of ubiquitin is firstly linked to a specific cysteine of the E1 enzyme by a thioester bond through ATP hydrolysis. Subsequently, the ubiquitin is transferred to a cysteine of the E2 enzyme [10], which then catalyzes the transfer of ubiquitin to the E3 enzyme and ultimately to the substrate protein (Figure 1). Such a process is common to the homologous to E6-AP carboxyl terminus (HECHT) ubiquitin ligase [10], as well as to the transfer of ubiquitin in close proximity of the catalytic subunit of a Really Interesting New Gene-finger (RING-finger) E3 ligase and the related U-Box domain to the substrate [11,12].

The ubiquitin bears different lysine residues, and each one of these can bind to another specific monomer of ubiquitin, giving rise ubiquitin chains with different topologies, such as K48 chains that lead to the proteasome targeting and degradation [13], or K63 chains that are involved in cell signaling events [14]. Furthermore, mixed or branched ubiquitin chains, N-terminal ubiquitylation, mono-ubiquitylation, and linear ubiquitylation can be generated as well (Figure 2) [15]. The majority of ubiquitin ligases belongs to a family of modular protein complex, called Cullin RING-finger Ligases (CRLs), which are composed of a scaffold subunit, named Cullin, a catalytic subunit named RING-box protein 1 (Rbx1), an adaptor protein and a substrate receptor [16]. The substrate receptor specifically binds to its own pool of selected target proteins, determining their ubiquitylation for proteasomal degradation or other possible destinations [17].

Pathogenic bacteria—such as *Salmonella typhimurium*, *Legionella*, and *Shigella*—have developed needle-like structures, namely the type-III secretion system (T3SS) and type-IV secretion system (T4SS), to deliver their E3 ubiquitin ligase effectors within the host cells [18,19]. Similarly to eukaryotic E3 proteins, the bacterial effector proteins are HECT and RING/U-Box ubiquitin ligases, but also codify for Novel E3 ligases (NEL) and new ligases [20], as well as for effectors that manipulate host E3s.

## 2. Ubiquitylation in Immune Response

The immune system represents the first defense line raised against pathogen invasion or more generally exogenous danger signals. It is mediated by pathogen-recognition receptors (PRRs) that defend us from invading pathogens by recognizing pathogen-associated molecular patterns (PAMPs) [21,22]. Ubiquitylation and its reversal, deubiquitylation, have been implicated in the development and regulation of the innate and adaptive immune responses. Increasing evidence supports their crucial role also in the orchestration of an immune response by ensuring the proper functioning of different cell types that collectively constitute the immune system [23]. In response to pathogens infection, the host innate immune system deploys a series of distinct antimicrobial activities—such as inflammatory signalling, phagosomal maturation, autophagy, and apoptosis—all of which are fine-tuned by the ubiquitin system with the ultimate goal to eradicate the invading pathogens and concomitantly to reduce host damage [24]. The innate immune system detects pathogens via the PRRs protein family. In particular, TLRs (Toll-like receptors), RLRs (Rig-I-like receptors), or NLRs (NOD-like receptors) signaling pathways, wherein, various E3-ligases have a critical role in ubiquitylation of key molecules [25]. TLRs and NLRs are directly responsible for the initiation and propagation of the inflammatory cascade in response to PAMPs. TLRs are tightly regulated by a number of E3 ubiquitin ligases—such as Peli1, TRAF3, and TRAF6—which drive the poly-ubiquitylation of several critical components mediating the pro-inflammatory response [26,27]. Tumor necrosis factor receptor (TNFR)-associated factors (TRAFs) constitute a family of proteins which includes six members (TRAF1-6) sharing peculiar motifs, including a homologous C-terminus domain termed TRAF domain and several zinc fingers domain. In addition, TRAF proteins, except TRAF1, contain a RING domain, at N-terminus, with E3 ubiquitin ligases activity. TRAFs proteins mediate signal transduction from a large variety of immune receptors, including TNF-receptors, and other cytokine receptors, including TGF-β receptors, pattern-recognition receptors (PRRs), and antigen receptors [28,29]. Within the TRAF family, TRAF6 is the better characterized E3 ligase that specifically conjugates lysine (K) 63-linked polyubiquitin chains [28]. Among the major downstream pathways regulated by TRAFs are those leading to activation of the nuclear factor κB (NF-κB) and mitogen-activated protein kinases (MAPKs), whose activity is in turn essential for the induction of genes associated with innate immunity, inflammation, and cell survival [27,30]. For instance, upon TLRs and IL-1R engagement, TRAF6 is recruited through the adaptor MyD88 and conjugates K63-linked ubiquitin chains onto itself as well as to other proteins, such as IRAK1 and the IKK regulatory subunit NEMO [31], leading to NF-kB activation and activating immune response.

TRAF4 and TRAF6 are also involved in the signaling of TGF-β receptor, triggered by TGF-β, which, if dysregulated can lead to induction of apoptosis, epithelial–mesenchymal transition, and cancer cell invasion [32]. Specifically, TGF-β activates TGFBR1, which recruits TRAF6 promoting its auto-ubiquitylation. TRAF6, in turn, triggers K63-polyubiquitylation of TAK1, leading to activation of p38 and JNK. Interestingly, TRAF6 could also promote the proteolytic cleavage of the polyubiquitylated TGFBR1 by TNF-alpha converting enzyme (TACE) and presenilin-1, which would release the intracellular domain of TGFBR1 for translocation to the nucleus, leading to the transcriptional activation of a specific subset of genes, including Snail and MMP2, ultimately promoting cancer cell invasiveness [32,33,34]. Noteworthy, TRAF4 is involved in the stabilization of TGFBRs on the plasma membrane, through the ubiquitylation and consequent degradation of SMURF2, an E3 ligase that acts as a negative regulator of TGF-β receptor [35]. Upon TNF-alpha stimulation, TNFR1 associates to TRADD, TRAF2, its homologue TRAF5, RIP1, and the E3 ubiquitin ligases cIAP1 and cIAP2, constituting a complex involved in the regulation of cell survival mechanisms. In this complex, cIAP1 and cIAP2 conjugates K63-linked ubiquitin chains to RIP1, which recruits and activates TAK1 and IKK kinase, leading to activation of NF-kB. Interestingly, TRAF3 acts as a negative regulator of the non-canonical NF-κB pathway, mediating the ubiquitin-dependent degradation of NIK [28,31].

Ubiquitylation is modulated also by a number of de-ubiquitylating enzymes (DUBs), specifically A20 (also known as TNFAIP3), which negatively regulates the TLR-induced response by ensuring the cleavage and removal of poly-ubiquitin chains from target proteins and thus terminate their signaling [26]. TLR signaling can lead, either directly or indirectly, to the induction of cell death through RIP1, RIP3, caspase-8, FADD, and cFLIP [23]. 

For instance, MyD88-dependent signaling, engaged by TLR4, induces the recruitment of IRAK proteins that subsequently interact with TRAF6, TRAF3, cIAP1, and cIAP2 [23]. Within this TLR4 signaling complex (TLR4-SC), TRAF6 is activated and interestingly, rather than the conventional Lys48 (K48)-linked poly-Ub chains that normally signal for protein degradation, it conjugates K63-linkages to itself and to cIAP1/2. In turn, cIAP1/2 serves as a scaffold for the recruitment of two kinase complexes, namely TAB-TAK and IKK-NEMO, leading to the activation of NF-kB and MAPK [23]. The TNF-signaling pathway is also involved in adaptive immune responses to pathogen invasion. In various immune cells, including T- and B-cells as well as macrophages, TNFα-induced NF-kB and apoptosis pathways are regulated by well-known Lys 63-linked ubiquitin signals and Met1-linked ubiquitin chains [36,37,38,39]. Lys 63-linked ubiquitin chains play a role upstream of this signaling pathway; cIAP ubiquitylates RIPK1 with Lys 63-linked chains that recruit the TAB2/TAK1 kinase complex and LUBAC, an E3 ligase complex composed of HOIP, HOIL-1L, and SHARPIN [40]. NLRs, NOD1 and NOD2 in particular, recognize bacterial peptidoglycan and autophagy receptors such as NDP52 and Optineurin (OPTN), can directly recognize invading bacteria [41]. RLRs, RIG-I, MDA5, and LGP2, are cytosolic PRR involved in sensing actively replicating double-stranded RNA (dsRNA) viruses [23]. In RIG-I signalling, a K11 poly-ubiquitylation of Beclin-1 (an inhibitor of RIG-I pathway) prevents its proteasomal degradation, which in turn further blocks RIG-I interaction with interferon-α promoter stimulator (IPS)-1 and inhibits the synthesis of type-I IFN [42]. 

In a DNA sensing innate-immune pathway, K27-polyubiquitination of cyclic GMP-AMP synthase (cGAS) and Stimulator of Interferon Genes (STING) protein are required for the activation of anti-viral innate-immune pathway [43]. Recently, it has been shown that the host cell-mediated ubiquitylation on *Salmonella* acts as a signaling scaffold to coordinate host cell-defense mechanisms via NF-κB pathway [44]. Specifically, M1-linked (linear) poly-ubiquitylation of the pathogen is catalyzed by the E3 ligase complex LUBAC [45] and removed by the de-ubiquitylase OTULIN [46], where LUBAC and OTULIN are ubiquitin chain-linkage specific enzymes that generate or degrade M1-linked ubiquitin chains, respectively [47]. 

Autophagy is an evolutionarily conserved process in which cells eliminate protein aggregates, damaged or redundant organelles, as well as bacterial pathogens from their cytosol or recycling cytosolic contents for metabolism [24,48]. The autophagic degradation of intracellular pathogens such as *Salmonella*, is termed ‘xenophagy’. During xenophagy, M1- or K63-linked ubiquitin chains present on bacteria are recognized by the UBAN domain of OPTN and TBK1-mediated phosphorylation of OPTN increases its binding affinity towards LC3, thus targeting the pathogen the autophagosome for degradation [44]. Ubiquitylation is not only involved in autophagy, but has a direct role also in regulating the inflammatory signaling pathways and the mechanisms of antigen presentation in adaptive immune responses. Sequestosome-1/p62-like receptors (SLRs), a family of innate immune receptors-a category of PRRs engaged in the recognition and capture of intracellular microbes, initiate autophagy to eliminate intracellular microbes by direct capture and delivery of antimicrobial peptides, and serve as an inflammatory signaling platform [43]. Several bacteria including *Shigella*, *Salmonella*, *Legionella*, and *Pseudomonas* induce activation of caspase-1 via the NLRC4 inflammasome in infected macrophages [49], inducing proteolytic maturation of caspase-1 and pro-inflammatory cytokines, such as IL-1β and IL-18, resulting in a form of cell death called pyroptosis [50]. *Shigella* induces rapid pyroptotic cell death, a prerequisite for the bacterial egress from macrophages. In particular, by delivering the IpaH7.8 E3 ubiquitin ligase effector and activating caspase-1 GLMN, a IpaH7-8 target, it specifically binds cellular inhibitor of apoptosis proteins 1 and 2 (cIAP1 and cIAP2), members of the inhibitor of apoptosis (IAP) family of RING-E3 ligases, which results in reduced E3 ligase activity, and consequently inflammasome-mediated cell death of macrophages [51].

## 3. HECT E3 Ubiquitin Ligases

As mentioned before, HECHT ubiquitin ligases bind covalently ubiquitin. Diverse pathogen effectors are able to mimic eukaryotic HECHT ubiquitin ligases. SopA is one of the firstly-identified HECH TSS3 effector in *Salmonella typhimurium* [52]. *Salmonella* invades host macrophages and epithelial cells, where it can survive and replicate in a peculiar structure, defined as *Salmonella* containing vacuole (SCV), eliciting a pro-inflammatory response [53]. At the molecular level, SopA exerts its function by ubiquitylating two host ubiquitin ligases, TRIM65 and TRIM56 (Figure 3A) [54], and thus blocking the anti-microbial response mediated by Interferon-α (IFN-α) [55] (Table 1). TSS3 effector NleL is another HECHT ubiquitin ligase from enterohemorrhagic *Escherichia coli* [56,57,58]. NleL is able to ubiquitylate and downregulate the Translocated intimin receptor (Tir), an effector secreted into host intestinal cells. In this scenario, NleL downregulates the formation of pedestals, which are host actin-built structures that this species of pathogen utilizes for embodying the intestinal epithelium [56,57]. Conversely, NleL is able to mono-ubiquitylate host c-Jun NH2-terminal kinases (JNK) at lysine 68, disrupting its interaction with the upstream kinase MKK7 and suppressing the activity of the transcription factor activator protein-1 (AP-1), which results in the formation of actin pedestals [58] (Table 1).

## 4. RING/U-Box Ubiquitin Ligases

Pathogenic RING/U-box ligases catalyze the transfer of ubiquitin from E2 directly to the substrate. Non-LEE-encoded (NleG) is a large family of 20 members U-box ubiquitin ligases type III effectors delivered by *Escherichia coli* [60] (Figure 3B) (Table 1). The C-terminal domain of NleG2-3 (residues 90 to 191) is the most conserved region of NleG proteins and was solved by Nuclear Magnetic Resonance (NMR) [60]. NleG proteins display a RING/U-box domain in the C-terminal region and the catalytic activity of NleG2-3 and NleG5-1 was demonstrated by in vitro ubiquitylation assay [60]. The screenings of 30 E2 enzymes showed that the NleG proteins preferentially interact with UBE2E2, UBE2E3, and UBE2D2 [60]. *Legionella pneumophila* translocates other RING-finger/U-box ligases. Among them, LubX E3 ligase possesses a double U-box domain that is located at the N-terminus and C-terminus of the protein (Table 1). Through the N-terminal U-box, LubX recruits the substrate and E2 enzymes—such as UBE2W, UBEL6, UBE2D, and UBE2E families [61]—and then undergoes a conformational change to bring the conjugating enzyme in close proximity with the C-terminal U-box, which ubiquitylates the host Cdc2-like kinase 1 (Clk1). However, the functional consequences of Clk1 ubiquitylation in this context are still not understood [62]. An important additional target of Lubx is the *Legionella* effector SidH, which makes possible to define Lubx as the “meta-effector” protein of *Legionella pneumophila* [63]. Indeed, LubX ligase targets SidH, which temporally accumulates in the host cell, for ubiquitylation and proteasomal degradation; by doing so, Lubx activity is able to attenuate a possible hyper-virulent phenotype of *Legionella* at a later stage of infection and to preserve the host survival [63]. SidC is an E3 ubiquitin ligase which has a role in membrane lipid bilayer modeling by binding to phosphatidylinositol 4-phosphate (PI_4_P), and is essential for the generation of the *Salmonella* containing vacuole (SCV), possibly by ubiquitylating proteins derived from the endoplasmic reticulum [59] (Table 1). *L. pneumophila* effector GobX has a central region homologous to the mammalian U-box motif and displays also ubiquitin ligase activity (Table 1). It represents a peculiar E3 ligase, as it undergoes palmitoylation and localizes within the Golgi complex. For this reason, GobX is able to exploit two different pathways of host cells: ubiquitylation and S-palmitoylation [64]. Regrettably, there is still no evidence supporting a role for GobX during infection. Likewise, a potential substrate of ubiquitylation remains elusive.

## 5. F-Box Proteins

The F-box proteins are substrate receptor of Skip/Cullin/F-box (SCF) complex, containing Cullin-1 [67]. Bacteria have developed a proper set of tools to hijack the Ubiquitin Proteasome System (UPS) of host cells. In particular, *Legionella pneumophila* has evolved a system to exploit the host Ubiquitin Proteasome System (UPS) by injecting the TSS4 effectors that modularly assemble within the host SCF complex [67] (Table 1). *Legionella pneumophila*-derived LegU1 and LegUA13 proteins, also known as AnkB and Ceg27, respectively, share a common ankirin domain at their C-terminus. LegU1 binds the SCF complex and degrades the host proteins BAT3 or BAG6 (Figure 3C), which are involved in the shuttling of newly synthetized tail-anchored (TA) proteins from the ribosome to the endoplasmic reticulum [65]. BAG6 is also involved in the protein quality control pathway, contributing to the proteasomal degradation of mislocalized proteins [66]. 

## 6. Novel E3 Ligases

Pathogens codify Novel E3 Ligases (NELs) that differ from the eukaryotic E3s. [68]. These ubiquitin ligases are constituted by an N-terminal Leucine-rich repeat (LLR) required for interaction with the substrate, and a catalytic C-terminus required for ubiquitylation of substrates [69]. The LLRs domains belong to the IpaH-SspH family and are conserved across numerous species of infectious bacteria, such as *Shigella flexneri*, *Salmonella enterica*, and some *Pseudomonas* species, including *Pseudomonas putida*, *Pseudomonas entomophila*, *Pseudomonas fluorescens*, and *Pseudomonas syringae*. The peculiarity of NEL ubiquitin ligases relays on the auto-inhibitory switch of their folding in absence of the specific substrate, due to the association of LRR domain to the catalytic cysteine present at their C-terminal [70,71]. Crystallography studies have showed how, once engaged to substrate, NEL ubiquitin ligases undergo a conformational switch, enabling the catalytic cysteine to ubiquitylate the substrate upon its recruitment by the LRR domain [70,71].

The IpaH family includes fifteen NEL ligases. *Shigella flexneri* encodes NEL IpaH ligases SspH1, SspH2, SlrP, and IpaH9.8 [72,73]. IpaH9.8 is secreted through the type three secretion system (TSS3) protein into the host cell, where it does translocate to the nucleus and participates to the splicing of U2AF35 mRNA through its E3 ligase activity; this action leads to repression of pro-inflammatory effects of cytokines upon infection [74]. 

In addition, IpaH9.8 targets for ubiquitylation and degradation other components of the NF-kB-dependent inflammatory response, including NEMO/IKKγ, which is a key regulator of NF-κB activation in response to specific stimuli [75]. Furthermore, IpaH9.8 ubiquitylates via Lys-48 human guanylate binding protein-1 (hGBP1), destining it for proteasomal degradation. hGBP1 belongs to a family of GBPs that possess the ability to penetrate the pathogen causing its death. Thus, hGBP1 possesses an antimicrobial activity [76]. IpaH0722 is an additional effector ligase that upon breakage of the infected vacuole is released into the cytoplasm in order to counteract the inflammation response through the inhibition of the NF-κB activation pathway. Indeed, IpaH0722 promotes the ubiquitylation coupled to proteasomal degradation of protein kinases C (PKCs) and TNF receptor-associated factor 2 (TRAF2), thus interfering with inflammation and immune system response [77]. On the same line, the IpaH4.5 ubiquitin ligase targets the p65 subunit of the NF-κB transcription factor for ubiquitylation and subsequent proteasome-dependent degradation, thus eliminating the most relevant transcriptional subunit of the NF-kB complex [77].

*Salmonella typhimurium* is another bacterial pathogen that delivers NEL ubiquitin ligases into the host cell, including the two Salmonella secreted protein H1 and H2 (SspH1, SspH2) and the Salmonella leucine-rich repeat protein (SlrP) [70]. SspH1 modulates the host inflammatory response by ubiquitylating the protein kinase N1 (PKN1) and destining it to proteasomal degradation. PKN1 is a serine/threonine kinase residing inside the nucleus, where it promotes the activation of androgen receptor, mineralocorticoid receptor, and progesterone receptor signaling; whereas it is an inhibitor of AKT activation and suppresses NF-kB activation. Its degradation, driven by SspH1 in the nucleus leads to downregulation of inflammation, activation of AKT, which is essential for *Salmonella* penetration inside the host cell and activation of NF-κB [75,77]. SspH2 undergoes S-palmitoylation at the host cell membrane and then interacts with NOD-like receptors SGT1 and nucleotide-binding oligomerization domain-containing protein 1 (NOD1), forming a trimeric complex. Thereafter, SspH2 mono-ubiquitylates and activates NOD1 eliciting the secretion of interleukin-8, which is fundamental in the innate immune system reaction [78]. Interleukin-8 is produced by blood cells and works as a chemoattractant for neutrophils at the site of infection [63]. Overall, it is reasonable that SspH1 and SspH2 might somehow regulate the host–pathogen balance between innate immune response and infection. SlrP ubiquitylates and determines the proteasome-mediated degradation of thioredoxin, causing the death of infected cells [79] as well as of the endoplasmic reticulum-resident chaperone protein ERdj3. Targeted degradation of ERdj3 implies a translocation of SlrP into the endoplasmic reticulum, where it may lead to an accumulation of unfolded proteins and eventually inducing ER stress, unfolded protein response (UPR) and ultimately triggering apoptosis of infected cells. [80].

Finally, it is worth to mention the atypical *Yersinia pseudotuberculosis* NEL E3 ligase YopM, which contains multiple LRR domains at the C-terminus and N-terminal ubiquitylating activity. The substrates of YopM are still not known. YopM acts as scaffold protein in the formation of a complex between ribosomal s6 kinase (RSK) and protein kinase R (PKR), whose role remains to be clarified [81]. YopM also inhibits caspase-1 in macrophages, which is important for the virulence of *Yersinia* [82]. 

## 7. New Effector Bacterial Ubiquitin Ligases

The process of ubiquitylation involving the *Legionella pneumophila* TSS4 effector SidE family members (SdeA, SdeB, SdeC, and SidE) utilises an alternative and peculiar way of substrate ubiquitylation that differs from the classical enzymatic cascade operating in eukaryotic cells. In fact, SidE proteins contain the mono-ADP-ribosyltransferase (mART) motif and activate ubiquitin by the transfer of ADP-ribose to the Arg42 of ubiquitin using NAD as cofactor. Subsequently, the ADP ribosylated-ubiquitin is transferred to Rab GTPase proteins of the endoplasmic reticulum with the release of AMP [83]. Recently, it has been shown that *Legionella pneumophila* uses SdeA molecule to remove Ub chains for its survival [84], as well as for ubiquitylating the host’s Rab-family GTPases via a novel phosphoribose linkage on serine residues [21,83,85,86]. SdeA, which belongs to the SidE effector family of Legionella pneumophila, can transfer ubiquitin to endoplasmic reticulum-associated Rab-family GTPases in a manner independent of E1 and E2 enzymes utilizes NAD+ as a cofactor to attach ubiquitin to a serine residue of the substrate. Kim and colleagues characterized the mono ADP-ribosyltransferase domain of SdeA and showed that consists of two sub-domains termed mART-N and mART-C [84].

Notably, this atypical first step within the ubiquitylation reaction described for the SidE system sheds light on a novel molecular mechanism that substantially diverges from the canonical pathway; nevertheless, it is still not well understood how these type of E1-E2 independent ubiquitylation of Rab GTPase may contribute to pathogenic virulence of *Legionella pneumophila* [83].

Finally, the effector protein RavN of *Legionella pneumophila* has been shown to belong to the growing class of bacterial proteins that mimic the host cell E3 ligases [87]. The E3 ligase activity is located within the N-terminal region of RavN and is dependent on the interaction with a defined subset of E2 ubiquitin-conjugating enzymes. The crystal structure of the N-terminal region of RavN revealed a U-box-like motif that is (albeit only remotely) similar to other U-box domains, indicating that RavN is evolutionary related to E3 ligase [87]. Moreover, Lin et al. have identified and experimentally validated four additional *Legionella pneumophila* effectors—namely Lpg2370, Lpg2577, Lpg2498, and Lpg2452—as paralogues of SidC [87]. 

## 8. Host Cell Ubiquitin Ligase Modulated by Bacterial Virulence Factors

Bacterial virulence factors do often act by modulating host ubiquitin system, inducing bacterial pathogenesis. Horvat et al. has reported how the *Helicobacter pylori* virulence factor CagA upregulated TRIP12 protein, an E3 ubiquitin ligase, inducing ubiquitylation and subsequent degradation of p14ARF, which preserve the host cell from the overt activation of p53-independent autophagy [88]. Interestingly, cycle inhibiting factors (Cif_s_) are type III secretion system effectors produced by some Gram-negative pathogenic bacteria, including enteropathogenic *Escherichia coli* (EPEC), enterohemorrhagic *Escherichia coli* (EHEC), *Yersinia pseudotuberculosis*, *Photorhabdus luminescens*, *Photorhabdus asymbiotica*, and *Burkholderia pseudomallei* [89]. The main targets of Cif_s_ in host cells appear to be Cullin RING E3 ubiquitin ligases (CRLs). Cif has been shown to catalyze the de-amidation to glutamate of a glutamine residue (Gln40) of NEDD8, thereby leading to inhibition of CRL-catalyzed poly-ubiquitin chain formation [89,90]. Inhibition of CRL activity by Cif resulted in the stabilization of CRL substrates including, among the others, the cell cycle inhibitors p21 and p27, ultimately leading to cell cycle arrest [90,91,92].

Moreover, McCormack et al. showed that a cullin-RING E3 ubiquitin ligase (CRL) complex, containing Cullin-1 and βTrCP mono-ubiquitylates Perforin-2. This counteracts the pathogens infection by causing a rapid redistribution of Perforin-2, which is essential for its bactericidal activity [93]. They also report how enteric pathogens, such as *Yersinia pseudotuberculosis* and enteropathogenic *Escherichia coli*, can disarm host cells defenses by injecting the cell cycle inhibiting factor (Cif) to induce the deamination the ubiquitin-like protein NEDD8. Because the CRL activity relies on NEDD8, Cif hampers the ubiquitin-dependent trafficking of Perforin-2, and, as a consequence, the host bactericidal activity [93].

Recently, the Listeriolysin O (LLO) toxin, secreted by *Listeria monocytogenes*, was shown to induce dramatic alterations of the host proteome, and specifically of several ubiquitin and ubiquitin-like ligases—including UBE2B, UBE2L3, UBE2T, UBE2V1, NEDD8, and UFM1—together with their respective E2 enzymes (UBC12/UBE2M and UFC1), resulting in a significant downregulation of their activity [94]. 

## 9. Conclusions

Protein ubiquitylation is a key post-translational modification process that can be involved in host–pathogen interaction and defense mechanisms. The ubiquitin pathway is a biochemical multistep cascade that relies on the sequential activity of activating (E1), conjugating (E2), and ubiquitin ligase (E3) enzymes. Bacteria lack of the ubiquitination machinery and have evolved molecules capable of mimicking some of the host ubiquitin ligases. Furthermore, they can exploit the target cells by modifying or deregulating key factors, in order to escape their immune surveillance, such as NF-kB and interferon pathways. The bacterial ubiquitin ligases are mostly transferred inside the target cells through the Type three and four (TSS3 and TSS4) secretion system, which are a needle-like structures of the bacteria that perforate the membrane and inject the effectors. Some other pathogens, such as *Salmonella Typhimrium*, *Legionella pneumophila*, and *Shigella flexneri*, invade the host cells by creating a vacuole containing bacterium. For this purpose, many effector ubiquitin ligases contribute to the formation of the vacuole by interacting with and hijacking the host machinery present at the cell membrane.

The bacterial ubiquitin ligases are classified as HECHT, RING-finger/U-box, and novel ligases. An atypical class of effector ubiquitin ligases use a mechanism of substrate ubiquitination that is different from the E1, E2, and E3 complexes canonical cascade and relies instead on the ADP ribosylation of ubiquitin. Furthermore, evidences suggest how virulence effectors released by many strains of pathogenic bacteria could also manipulate host ubiquitin ligases.

Altogether, in a context (such as in prokaryotic cells) lacking the ubiquitin system, the development of E3 ligase for taking advantage of the eukaryotic ubiquitin pathway does represent an extraordinary example of convergent evolution, in which two living organisms encounter each other and indeed evolve toward the selection and optimization (albeit for antithetic purposes) of the same function.

A deeper understanding of the mechanisms dictating ubiquitin exploitation, involving canonical, non-canonical ubiquitylation, de-ubiquitylation, or a specific ubiquitylation code, could be harnessed for developing novel antimicrobial therapeutic agents, such as novel classes of antibiotics, in order to overcome one of the most difficult challenge currently faced in clinics, namely the emergence of pathogenic strains resistant to conventional antibiotics. As an example, the IpaH family, which is characterized by a high grade of conservation among Gram-negative bacteria and is involved in the dysregulation of key factors of immune response, including the TNF-α and NF-kB pathways, could represent a suitable candidate for the development of novel drug inhibitors, with a broad applicability [69], in view of their presence in almost all Gram-negative bacteria and of their highly specific mechanism of action, which is distinct from the one operated by mammalian E3 ligases. However, the task of targeting E3 ubiquitin ligases as therapeutic approach is made more challenging by the lack of ATP binding pockets. As such, this feature limits the possibility to deploy ATP-analogues to inhibit catalytic activity, an approach that has already been used and proved successful for the design and/or identification of protein kinase inhibitors [95]. Thus, due to the absence of ATP-binding sites within the RING and HECT ubiquitin ligase domains, the real challenge is the identification of a drug targeting protein–protein interactions between either the ubiquitin ligases and their specific substrates or to the upstream E2-conjugating enzymes. To date, a successful example is represented by the chemical compound oxathiazol-2-one, which preferentially inhibits the proteasome of *Mycobacterium tuberculosis*, rather than the mammalian counterpart [95]. Regrettably, *Mycobacterium tuberculosis* is the only bacterial pathogen known to have a proteasome, which limits the application of this drug to treat specifically mycobacterial infections.

Although host mimicry is a common strategy utilized by pathogens that hijack specific cell function and/or imitate host motifs [96], the development of antimicrobial agents targeting structural mimics is not fully efficient due to their similarity to host factors. An interesting case is represented by the NIeG effector family of E3 ubiquitin ligases from enterohemorrhagic *E. coli*, which shows a high grade of structural and functional similarity with eukaryotic E3 enzymes, despite the lack of any homology in terms of amino-acidic sequence [97]. Interestingly, in this setting, pathogen effectors can be targeted by host modification systems, such as ubiquitylation or SUMOylation.

It is also interesting to note how, under some specific circumstances, the activity of the pathogen effectors can be regulated by a host-depending post-translational modification—thus restricting the function of pathogen protein as a function of the host environment. Specifically, the activity of Salmonella effector SopB is regulated by a specific mono-ubiquitylation at the plasma membrane [98]. These effectors may be represent an useful tool to understand the biology of host cell, their responses to infection and eventually pave the way to develop novel therapeutic strategies [99]. As described in the previous paragraphs, the ability to modulate the eukaryotic ubiquitylation system is a hallmark feature of several intracellular pathogens, including Legionella, Shigella, and EHEC/EPEC [100,101,102], providing novel targets to develop antibiotic strategies, in an era characterized by an extremely worrying increase of phenomena of antibiotic resistance. Thus, it is of fundamental importance to gain further understandings of the mechanisms operated by bacterial effectors in order to hijack host ubiquitylation pathways. For instance, among the still so many unanswered questions, identifying whether the mono-ubiquitylation of Salmonella SopB is performed by host or bacterial E3 ligases, or which E3 ligase is responsible for the apposition of K63-linked ubiquitin chains on the Salmonella outer membrane [103] and which pathogen effector targets remain to be identified [104], and could help developing novel anti-bacterial therapeutic strategies besides the common and well-established ones based on traditional antibiotics.

## Figures and Tables

**Figure 1 ijms-19-03576-f001:**
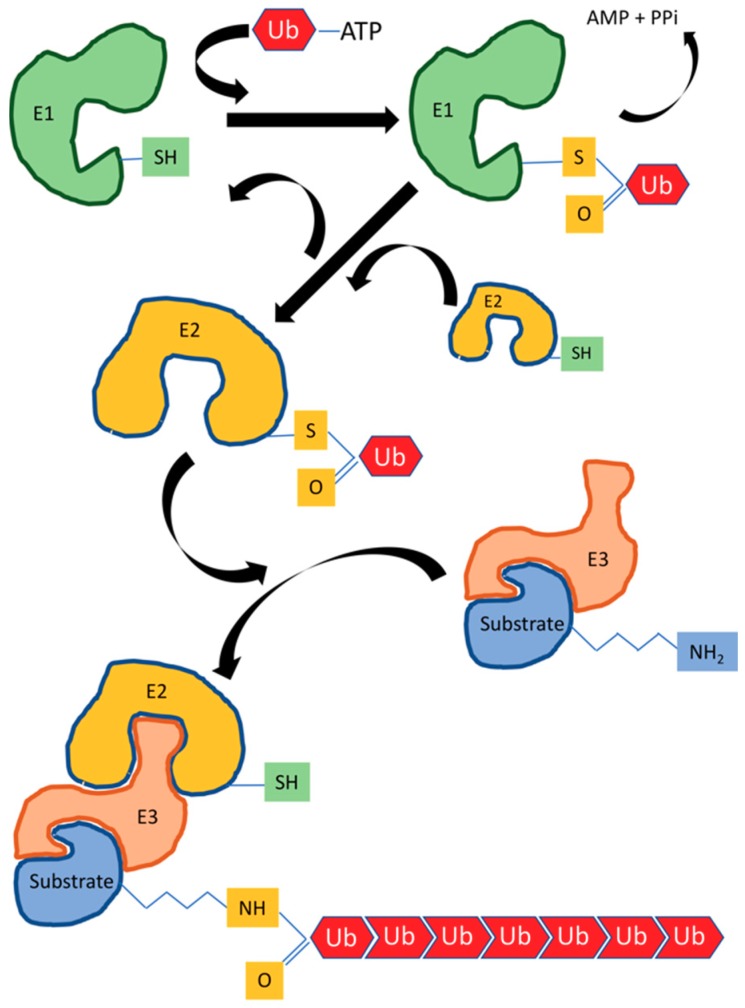
The sequential steps of ubiquitylation reaction. Three enzymes work sequentially in the reactions cascade of ubiquitylation: E1, the activating enzyme; E2, the conjugating enzyme; and E3, the ligase enzyme. The glycine carboxy-terminal of ubiquitin is firstly linked to a specific cysteine of the E1 enzyme by a thioester bond through ATP hydrolysis, and subsequently transferred to a cysteine of the E2 enzyme, which in turn catalyze the transfer of ubiquitin to the E3 enzyme and eventually to protein substrate.

**Figure 2 ijms-19-03576-f002:**
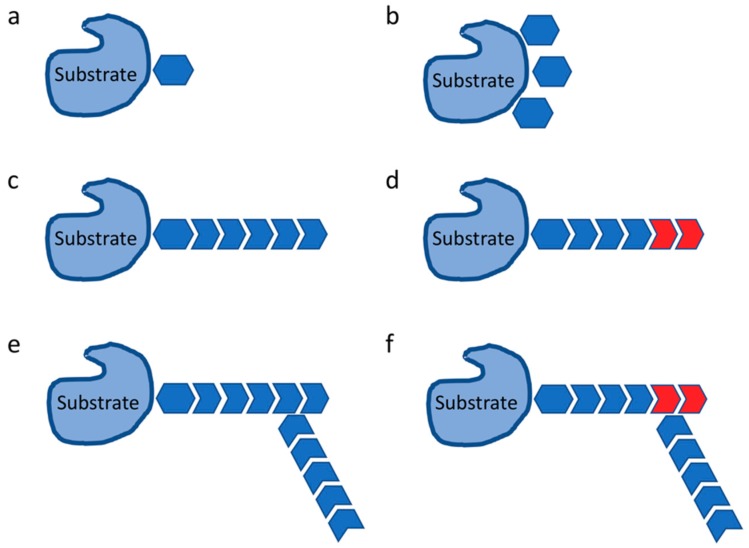
Ubiquitylation signals. Depending on the intracellular pathway the substrate proteins are part of/involved into, as well as on their final outcome (i.e., protein degradation and turn-over vs. change in localization, differential interaction, etc.), several types of ubiquitylation signals can be generated, such as mono-ubiquitylation signals, bearing (on the substrate protein) either a single (**a**) or multiple (**b**) ubiquitin monomers. Poly-ubiquitylation signals can determine more complex moieties, leading to homotypic (**c**), mixed (**d**), branched (**e**), or branched and mixed chains (**f**) of ubiquitin.

**Figure 3 ijms-19-03576-f003:**
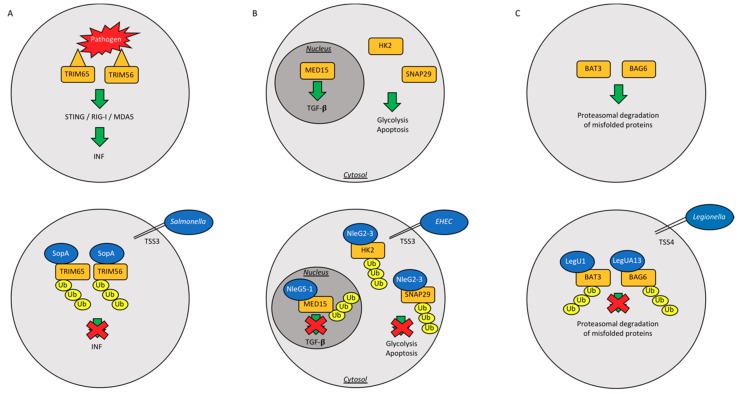
Paradigmatic examples of bacterial exploiting strategies. Schematic drawings depicting the three major and most common exploiting strategies adopted by bacteria to: infect host cells, escape their immune response and neutralize the activation of apoptotic/cell death pathways, and ultimately allow a full a pathogen propagation. In particular, the physiological pathways that are targeted by *Salmonella* (**A**), enterohemorrhagic *Escherichia Coli* (**B**), and *Legionella* (**C**), respectively, are reported in the upper panels. Human proteins are showed in yellow squares, whereas bacterial proteins are showed in blue circles.

**Table 1 ijms-19-03576-t001:** Main strategies executed by commonly and clinically relevant pathogenic bacterial strain to hijack cellular system and evade the host surveillance system.

Bacterial Strain	Bacterial Protein	Effector Type	Action	Reference
*Salmonella typhimurium*	SopA	HECH TSS3	Ubiquitinates host TRIM65 and TRIM56, blocking the production of interferon-α, helps machrophages and epithelial cell invasion	[52,53,54,55]
	SidC	RING/U-Box E3	Binds membrane PI4P and helps SCV generation	[59]
*Enterohemorrhagic Escherichia coli*	NleL	HECH TSS3	Ubiquitinates Tir into host intestinal cells; Ubiquitinates JNK	[56,57,58]
*Escherichia coli*	NleG	U-box Ubiquitin Ligase	Interacts with UBE2E2, UBE2E3 and UBE2D2	[60]
*Legionella pneumophila*	LubX	U-box Ubiquitin Ligase	Ubiquitinates Clk1; ubiquitinates SidH to preserve host survival	[61,62,63]
	GobX	U-box Ubiquitin Ligase	Exploites host ubiquitination and palmitoylation	[64]
	LegU1/LegUA13	F-box proteins and TSS4	Degrades host BAT3 or BAG6	[65,66]

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
