# Peer review of "Revisiting Bacterial Ubiquitin Ligase Effectors: Weapons for Host Exploitation"

_ijms, 2018, doi:10.3390/ijms19113576_

Round 1

Reviewer 1 Report

Two comprehensive reviews on bacterial E3 ligase effectors exploiting host ubiquitin systems have been published recently (Ashida and Sasakawa, 2017; Maculins et al., 2016). This reviewer suggests the authors could shift the focus to the latest findings and make the manuscript more concise. Also it would be much straightforward if the authors could summarize bacterial E3 ligase targeted signaling pathways with cartoon.

Author Response

Point to point response to reviewer

Reviewer  1

Two comprehensive reviews on bacterial E3 ligase effectors exploiting host ubiquitin systems have been published recently (Ashida and Sasakawa, 2017; Maculins et al., 2016). This reviewer suggests the authors could shift the focus to the latest findings and make the manuscript more concise. Also it would be much straightforward if the authors could summarize bacterial E3 ligase targeted signaling pathways with cartoon.

Response to Reviewer  1

We are thankful to Reviewer 1 for his/her suggestions that have been all taken in account. According to Reviewer 1, we have implemented the manuscript with Figure 3, which it includes now  a summary of bacterial E3 ligase targeted signaling pathways with cartoon.

Reviewer 2 Report

The review article 'Revisiting Bacterial Ubiquitin Ligase Effectors: Weapons for Host Exploitation' is interesting and is of potential importance to the research field.

Some commnets to improve the quality of the review.

1. The authors mentioned Lysine is the acceptor lysine. Is it known if other amino acids are involved in the process of ubiquitination.

2. The authors mentioned the role of the E3 ubiquitin ligase TRAF2, TRAF family members like TRAF2 and TRAF6 has emerged as an important  drivers related to Inflammation, Cancer and more recently the proteolytic shedding of the TGFbeta receptors. The authors could shed light on these aspects.

3. Can the ubiquitination mechanism be harnessed for developing therapeutic agents, The authors could give brief insights about the recent developments in the field.

4. The conclusion is quite short. would be good to describe the current state of the field and the future directions of the field.

Author Response

Point to point response to reviewers

Reviewer  2

The review article 'Revisiting Bacterial Ubiquitin Ligase Effectors: Weapons for Host Exploitation' is interesting and is of potential importance to the research field.

Some comments to improve the quality of the review.

1. The authors mentioned Lysine is the acceptor lysine. Is it known if other amino acids are involved in the process of ubiquitination.

2. The authors mentioned the role of the E3 ubiquitin ligase TRAF2, TRAF family members like TRAF2 and TRAF6 has emerged as an important  drivers related to Inflammation, Cancer and more recently the proteolytic shedding of the TGFbeta receptors. The authors could shed light on these aspects.

3. Can the ubiquitination mechanism be harnessed for developing therapeutic agents, The authors could give brief insights about the recent developments in the field.

4. The conclusion is quite short. would be good to describe the current state of the field and the future directions of the field.

Response to Reviewer  2

We are thankful to Reviewer 1 for his/her suggestions that have been all taken in account.

Point 1. In “Introduction” section, at pages 1-2 , we have discussed additional mechanisms of ubiquitylation involving non canonical amino acids in ubiquitylation process.

Point 2. In “Ubiquitylation in Immune Response” section, at pages 3-4, we have discussed the roles of TRAF members in inflammation, cancer and in the cleavage  of TGFbeta receptor.

Point 3. In “Conclusions” section, at pages 11-13, we have discussed the developing of therapeutic agents through the harnessing of ubiquitination mechanism.

Point 4. According to reviewer’s suggestion, we have strengthened the conclusions (pag. 11-13).

Round 2

Reviewer 1 Report

The authors have revised the manuscript according to the review's suggestion. This reviewer agrees the manuscript to be published.